# Design and Research of Inductive Oil Pollutant Detection Sensor Based on High Gradient Magnetic Field Structure

**DOI:** 10.3390/mi12060638

**Published:** 2021-05-30

**Authors:** Wei Li, Chenzhao Bai, Chengjie Wang, Hongpeng Zhang, Lebile Ilerioluwa, Xiaotian Wang, Shuang Yu, Guobin Li

**Affiliations:** School of Marine Engineering, Dalian Maritime University, Dalian 116026, China; dmuliwei@dlmu.edu.cn (W.L.); baichenz@dlmu.edu.cn (C.B.); wangcj@dlmu.edu.cn (C.W.); lebilei7@gmail.com (L.I.); wxtloveslife@dlmu.edu.cn (X.W.); yush@dlmu.edu.cn (S.Y.); liguobin@dlmu.edu.cn (G.L.)

**Keywords:** high-gradient magnetic field, planar coil, inductance detection sensor, oil contaminant detection

## Abstract

An inductive oil pollutant detection sensor based on a high-gradient magnetic field structure is designed in this paper, which is mainly used for online detection and fault analysis of pollutants in hydraulic and lubricating oil systems. The innovation of the sensor is based on the inductance detection method. Permalloy is embedded in the sensing region of the sensor, so that the detection area generates a high gradient magnetic field to enhance the detection accuracy of the sensor. Compared with traditional inductive sensors, the sensor has a significant improvement in detection accuracy, and the addition of permalloy greatly improves the stability of the sensor’s detection unit structure. The article theoretically analyzes the working principle of the sensor, optimizes the design parameters and structure of the sensor through simulation, determines the best permalloy parameters, and establishes an experimental system for verification. Experimental results show that when a piece of permalloy is added to the sensing unit, the signal-to-noise ratio (SNR) of iron particles is increased by more than 20%, and the signal-to-noise ratio of copper particles is increased by more than 70%. When two pieces of permalloy are added, the signal-to-noise ratio for iron particles is increased by more than 70%, and the SNR for copper particles is increased several times. This method raises the lower limit of detection for ferromagnetic metal particles to 20 μm, and the lower limit for detection of non-ferromagnetic metal particles to 80 μm, which is the higher detection accuracy of the planar coil sensors. This paper provides a new and faster online method for pollutant detection in oil, which is of great significance for diagnosing and monitoring the health of oil in mechanical systems.

## 1. Introduction

Condition monitoring and fault diagnosis of machinery are important in modern industrial contexts. Many crucial industries, including aerospace, offshore platforms, smart ships, and wind power generation, urgently need faster and more accurate monitoring technology to improve the productivity of their machineries. Online monitoring of machinery health can effectively reduce the need for sudden shutdown of vital machinery which can lead to loss of income, significantly reducing operating costs. Recently, many researchers are looking for methods that can accurately identify the state of machineries, realize remote monitoring, and predict inspection and maintenance arrangements in advance [1]. Although ferrographic analysis, vibration analysis, and thermal imaging technology have long been applied to wear particle identification and diagnosis, they are affected by factors such as equipment and environment, and have great limitations in online wear particle monitoring. Lubricating oil analysis technology has become an effective means of early warning about equipment failures, equipment wear, and aging status because oil serves as the “blood” of mechanical equipment [2]. Wear particles and external contaminants do exist in lubricating oil and circulate in the mechanical system together with the oil [3]. Under normal operating conditions of the system, the size of solid particles is usually 10–20 μm. When the mechanical components are abnormally worn, the abrasive grain size can reach 50–100 μm [4,5,6,7,8] or more, which will affect the normal operation of the equipment.

For the identification of metal abrasive contaminants in oil, inductive sensors based on 3-D solenoids and planar coils are commonly used. The inductive wear particle analysis method is less affected by environmental factors, can effectively distinguish ferromagnetic and non-ferromagnetic metals, and it is accepted by the majority of researchers. Based on the 3-D solenoid development technology, Du et al. [9] designed a 3-D solenoid sensor to detect 55 μm iron particles and 100 μm copper particles by using an LCR precision meter. Xiao et al. [10] designed an inductive sensor with a gradient magnetic production structure. This structure uses a large excitation coil to provide a gradient magnetic field. The gradient magnetic field passes through a small crack to form a detection area. This structure can increase the sample flux. Ren et al. [11] designed a three-coil inductive structure. The sensor consists of a detection coil and an excitation coil on both sides of the detection coil. The excitation coil provides the same excitation magnetic field, and the middle coil is responsible for detection. The sensor can detect 130 μm ferromagnetic and 230 μm non-ferromagnetic particles. However, due to the large range of the 3-D solenoid sensing area, when detecting multiple low-sized abrasive particles at the same time, they are likely to be identified as one large abrasive particle, which may cause possible false alarms of machine failures.

In order to solve the above-mentioned problems, planar coil sensors have been developed. In the case of the same number of turns, the sensing area of the planar coil is smaller, and the detection accuracy is better than that of a 3-D solenoid. Du et al. [12] designed a resonant high-sensitivity detection unit with two-layer planar coils, which can effectively identify 50 μm iron particles. Zeng et al. [13] designed a double-planar coil structure. Through the connection of two planar coil circuits, it can effectively detect four types of pollutants such as ferromagnetic, non-ferromagnetic particles, water droplets, and bubbles, but the detection accuracy is low. Shi et al. [14] designed a high-gradient magnetic field sensor based on the structure of the double-coil, which improved the detection accuracy of the double-planar coil and was able to identify 30 μm iron particles and 100 μm copper particles. The double-coil structure used in the above detection method mainly uses the mutual inductance generated by the double-coil which is greater than the inductance value of a single coil. However, this method is not stable because the double coils are not completely consistent during the production and installation process, and the mutual inductance value generated is also lower than the theoretical value. In comparison, the single-plane coil has better structural stability, and the production and installation of the sensing unit has less influence on the detection accuracy. Zhang et al. [15] used a single-layer planar coil to design a sensing unit and a detection system, which verified the stability of the single-layer planar coil structure. Ma et al. [16] added an iron core to the inner hole of the single-layer planar coil, and used the high magnetic permeability of the iron core to increase the magnetic field strength of the sensor area, greatly improving the detection accuracy of the single-layer planar coil, allowing it to identify 50 μm ferromagnetic and 130 μm non-ferromagnetic particles.

There is still a lot of room for improvement in the detection accuracy of the single-layer planar coil structure. Therefore, this paper proposes a design for a gradient magnetic field structure based on the single-layer planar coil. Embedded high-permeability material-permalloy on both sides of the plane coil. Permalloy is an alloy material with super high magnetic permeability, which is widely used in industry and sensor fields. The design of the new structure can effectively improve the magnetic induction intensity of the detection magnetic field, enhance the detection accuracy of the planar coil structure, and weaken the interference of the external magnetic field, so that the coil magnetic field is concentrated in the detection area.

## 2. Sensor Design and Manufacturing Process

The overall design of the sensor is shown in Figure 1a,c. The microchannel has a diameter of 300 μm, passing through the inner hole of the plane coil and two permalloy notches. The diameter of the enameled wire of the coil is 100 μm, and the number of turns is 30. Permalloy is 8 mm long, 2 mm high, and 0.3 mm wide. It is fixed on both ends of the coil, which is tightly attached in parallel to the coil. The core detection area of the sensor is shown in Figure 1b. The position of the microchannel and the permalloy notch parameters will be analyzed in the following content.

The manufacturing process of the permalloy is as follows (Figure 1d). The permalloy material was first prepared using a precision grinding wheel cutting machine (TWC50-A, Shen Zhen He Dong Technology Co., Ltd., Shen Zhen, China) to cut out different sizes of permalloy. It was then placed in a hydrogen furnace (QSXF-2-12, Hang Zhou Lan Tu Precision Instruments Co., Ltd., Hang Zhou, China) for annealing treatment. The heat-treated permalloy has a strong magnetic permeability. A precision winding machine (Shi Li SRDZ23-1B, Zhong Shan Shi Li Wire Winder Equipment, Zhong Shan, China) was used to wind the plane coil, and fix the wound coil on the correct position on the silicon plate. The permalloy was then fixed on both sides of the coil, a 7 cm long copper rod with a diameter of 300 μm was used as the mold for making the microchannel, and the mold was inserted into the permalloy slot and the inner hole of the plane coil. After mixing PDMS (polydimethylsiloxane) and curing agent 10:1, the model was casted with the mixture, and the copper rod for making the microchannel was drawn out, encapsulated, and the sample inlet and outlet were made.

## 3. Detection Principle and Simulation Analysis

When an alternating current is applied to the coil, Biot–Savart’s law shows that this type of coil can be simplified as the magnetic field of a circular current-carrying wire, and the magnetic field on its axis is shown in Figure 2. When point P is located at the center of the circle, the magnetic induction intensity at the center of the coil is the largest, and the distance r is the minimum circumference radius R, then the value of point P is:(1)$$B=\frac{\mu_{0}I}{2R},$$

For a symmetrical magnetic field, when only oil flows through the microchannel, assuming that the magnetic permeability in the coil is the vacuum permeability $u_{0}$, the magnetic induction intensity is $B_{0}$, and the relationship between the magnetic field intensity H and the magnetic induction intensity $B_{0}$ is:(2)$$B_{0}=\mu_{0}H,$$

When metal particles pass through this position, an eddy current effect will be generated at this position, which reduces the magnetic induction intensity of the original magnetic field. Let the reduced magnetic flux density be $B_{1}$. When a spherical metal particle with a radius of $r_{0}$ passes through the center of the coil, the magnetic induction intensity $B_{2}$ at that position is:(3)$$B_{2}=\mu_{0}\mu_{r}H,$$

$u_{r}$ is the relative permeability of metal particles. The relationship between the magnetic flux ∅ and the magnetic induction intensity B is ∅=BS, where S is the cross-sectional area of the metal particle at this position, and the magnetic flux at this position is:(4)$$\Phi_{1}=\left(B_{2}-B_{1}\right)S=\mu_{0}\mu_{r}HS-B_{1}S,$$

The inductance of the coil L is:(5)$$L=\Phi /I_{0},$$

$I_{0}$ for ferromagnetic particles, the relative permeability is much greater than the eddy current effect produced by vacuum permeability and particles, so when ferromagnetic particles pass through, the inductance of the coil will increase. For non-ferromagnetic metal particles, the relative permeability is approximately equal to the vacuum permeability, so when the non-ferromagnetic metal particles pass by, the inductance of the coil will decrease. It can be seen from Equation (4) that the larger the volume of the particle, the larger the cross-sectional area S at that position, and the greater the inductance change value produced, so the inductance change value is proportional to the volume of the particle.

This paper is based on the electromagnetic model of the finite element software COMSOL (COMSOL Multiphysics 5.3, COMSOL Inc., Stockholm, Sweden), and the solution variable is magnetic loss ***A***. Without considering the displacement current:(6)∇×A=B,

Take the curl of both sides:(7)∇×(∇×A)=∇×B,

Using mathematical formulas:(8)$$\nabla (\nabla \cdot A)-\nabla^{2}A=\mu \nabla \times H,$$
(9)$$\nabla \times H=J=\sigma E=\sigma \left(-\frac{\partial A}{\partial t}-\nabla V\right),$$
(10)$$\nabla^{2}A=\sigma \mu (\frac{\partial A}{\partial t}-\nabla V)-\frac{1}{\mu }\left(\nabla \mu \right)\times \left(\nabla A\right),$$

The governing equation of the detected part (particle, coil and air between coil and particle) can be obtained from Equation (10).

Particle interior:(11)$$\nabla^{2}A=\sigma \mu \frac{\partial A}{\partial t},$$

Air part:(12)$$\nabla^{2}A=0,$$

Coil part:(13)$$\nabla^{2}A=-\mu J,$$

After high-frequency alternating current is applied to the coil, the magnetic field inside and outside the particle changes due to the action of magnetization and eddy current, which in turn is captured by the current-carrying coil, thus causing the impedance change of the inductor coil.

The following was the process to simulate and analyze the coupling magnetic field of the coil and the permalloy. First, a single piece of permalloy is simulated. The simulation tool selects the electromagnetic field module in the COMSOL software. The finite element simulation parameters are the same as the experimental settings. The number of turns of the coil is 30 turns, the diameter of the enameled wire is 100 μm, and it is connected to 2 V alternating current and 2 MHz frequency. The size of Permalloy is the same as the sensor settings, and the simulation is proportional.

As shown in Figure 3a, triangular grooves with different angles are designed at the center of Permalloy, the groove depth is 1 mm, and the angles are 45°, 60°, 90°, 120°, and 150°. The corresponding simulation result picture is shown in Figure 3b–f. When the groove depth is constant, the smaller the opening angle, the greater the magnetic flux density. As shown in Figure 3b when the permalloy groove angle is at 45° the maximum magnetic flux density is 8.38 μT, and the maximum position is near the edge of the coil. As the angle increases, the magnetic flux density gradually decreases. When the angle is 150° as in Figure 3f, the maximum magnetic flux density is 1.34 μT, near the edge of the coil. The simulation result curves of different angles are shown in Figure 4a.

Figure 5 shows the simulation results of the permalloy rectangular groove. The depth of the rectangular groove shown in Figure 5a is 1 mm, and the groove width is 300 μm, 600 μm, 900 μm and 1200 μm. The corresponding simulation result picture is shown in Figure 3b–e. Different from the simulation result of the triangular groove, the magnetic flux density of the permalloy of the rectangular groove structure is much greater than that of the permalloy of the triangular structure after being magnetized. When the width of the rectangle is 300 μm, the maximum magnetic flux density is 14.81 μT; When the width of the rectangle is 900 μm, the maximum magnetic flux density is 40.80 μT, which is also the largest magnetic flux density among all the results. The result curve is shown in Figure 4b.

According to the analysis of the above simulation results, when the rectangular slot width is 900 μm, the simulation effect is the best, and the maximum value is at the rectangular slot near the edge of the coil, as shown in Figure 6a. When the edge of the rectangular slot is closer to the edge of the inner hole of the coil, the magnetized permalloy magnetic flux is larger, and the magnetic field of the detection area is stronger. Therefore, we continue to optimize the rectangular slot. As shown in Figure 6b, the rectangular slot has a width of 300 μm and a depth of 500 μm, and it is close to the edge of the inner hole of the plane coil. The related simulations on these structures were carried out, and the results are shown in Figure 6c,d. In the position shown in Figure 6b, the maximum magnetic flux density is 76.72 μT, and the degree of increase in magnetic flux is much greater than the magnetic flux density at the position in Figure 6a. Therefore, the rectangular groove shown in Figure 6b is selected as the best position of permalloy. Figure 6c is the simulation result corresponding to Figure 6b, and Figure 6d is the result of two permalloys of the same specification. Figure 6d shows that a piece of the same gauge is added on the basis of a single piece of permalloy. At this time, the magnetic flux density inside the coil is also doubled. After simulation analysis, the structure in Figure 6d was chosen since it gives the optimal result, and the subsequent experimental verification is also based on this conclusion.

Figure 7 shows the magnetic field distribution in different directions in the sensing region. Figure 7a is the horizontal magnetic field distribution. Compared with V-shaped groove permalloy, the rectangular groove permalloy can significantly enhance the magnetic field in the sensing region, and the magnetic field intensity at both ends of the induction region near the coil is the highest. Figure 7b is the horizontal magnetic field distribution. With the increase of height, the magnetic field intensity gradually decreases, and the effect of the same rectangular groove permalloy is better than that of V-groove permalloy in enhancing the magnetic field in the sensing area.

## 4. Experiment and Data Analysis

The experimental system consisted of an impedance analyzer (Keysight E4980A, Agilent Technologies Inc., Bayan Lepas, Malaysia), a microscope (Nikon AZ100, Nikon, Tokyo, Japan), a micro-injection pump (Harvard Apparatus B-85259, Harvard Apparatus, Holliston, MA, USA), a computer with a LabVIEW data acquisition unit and our high-gradient magnetic field sensor. The iron particles and copper particles used in the experiment were customized by professional laboratory equipment manufacturers (Beijing Huafeng Experimental Materials Co., Ltd., Beijing, China). The test bench system is shown in Figure 8.

Before the experiment, the mixture of oil and metal particles was first prepared. Iron particles with sizes of 20 μm, 30 μm, 40 μm, 50 μm, 60 μm and 70 μm were used. A total of 5 mg was weighed with a precision balance, then mixed with 120 mL of hydraulic oil. The mixture was placed on an ultrasonic oscillator (IKA S25, IKA Inc., Staufen, Germany) and shaken well for 2 min. The non-ferromagnetic particles were represented by copper particles. Copper particles of 80 μm, 90 μm, 100 μm, 110 μm, 120 μm and 130 μm were prepared, 6 mg was weighed with a precision balance, and then mixed with 120 mL hydraulic oil. Similar to the iron particles, the mixture of copper particle and oil was placed on an ultrasonic oscillator and shaken for 2 min. Finally, samples of different specifications of particles and oil mixtures were taken out and placed in the micro-syringe pump. The flow rate of the micro-syringe pump was adjusted to 50 μL/min, the impedance analyzer output voltage was set to 2 V, and the output frequency was 2 MHz (the highest frequency provided by Keysight E4980A is 2 MHz).

All experiments are the results of multiple measurements in dynamic mode. The experiment adopts the rectangular structure in Figure 6e. The comparison experiment was carried out using sensor with no permalloy, one piece of permalloy, and two pieces of permalloy. The iron particles of 70 μm and 100 μm, and the copper particles of 140 μm and 150 μm were randomly selected to compare the detection signals. The signal results are shown in Figure 9. The magnetization of iron particles is greater than that of eddy currents, so upward signals were generated, as shown in Figure 9a,b; the magnetization of copper particles is smaller than that of eddy currents, so downward signals were generated, such as Figure 9c,d.

In Figure 9, the red curve is the signal diagram of particles without permalloy; the blue is the detection signal diagram of particles with only one piece of permalloy; the black curve is the detection signal diagram of particles with two pieces of permalloy. The result of the detection signal in the Figure 8 reflects that after adding permalloy at both ends of the coil, the value of the detection signal is significantly improved. For 140 μm copper particles, when there is no permalloy, the sensor cannot capture the detection signal of copper particles; when a piece of permalloy is added, it can detect the signal of 140 μm copper particles, and the average signal value at this time is −7.2045 × 10^−10^ H; when two pieces of permalloy are added, the average detection signal value is −2.1850 × 10^−9^ H.

The signal-to-noise ratio (SNR) expresses the detection capability of the sensor. The larger the value of the SNR, the stronger the detection effect, and the smaller the value of SNR, the worse the detection effect. The formula for calculating the SNR is: (14)$$\;\mathrm{SNR}=\frac{\mathrm{Signal}\;\mathrm{value}}{\mathrm{Noise}\;\mathrm{value}}$$

Signal value is the maximum value of the signal minus the average noise value. Noise value equals average maximum minus average minimum. As shown in Figure 10.

In order to further verify the ability of permalloy to improve the detection accuracy of the sensor, the experimental group tested different sizes of iron particles and copper particles. The signal values and signal-to-noise ratio (SNR) values obtained from the experiments are shown in Figure 11 and Figure 12.

Figure 11a shows the average signal value of different iron particles, which confirms the previous conclusion. When permalloy is added to the sensor, the amplitude of the detection signal significantly increases, and the larger the particle size, the more obvious the signal increase. Figure 11b shows the SNR values of iron particles of different sizes. The SNR ratio reflects the accuracy of the sensor for particle detection. In Figure 11b, when the iron particle size is 30 μm and 40 μm, the SNR of the sensor without permalloy is 1, which indicates that the sensor cannot detect particles of this size. The sensor with a piece of permalloy has a SNR of 1 for detecting iron particles of 30 μm, and the sensor cannot detect particles of 30 μm. In contrast, the sensor with two pieces of permalloy has a SNR of 1.47 when detecting iron particles of 30 μm, which indicates that the sensor can detect iron particles below 30 μm. As the particle size increases, the SNR value increases, and the SNR of the sensor with two pieces of permalloy drastically increases as compared to others. Figure 12 shows the detection signal value and SNR of copper particles. The result of Figure 12a shows that the signal value of the sensor without permalloy is lower than that of the sensor with permalloy when detecting copper particles of different sizes. The lower detection limit of the sensor without permalloy is 150 μm, the sensor with one piece of permalloy can detect 130 μm, and the sensor with two pieces of permalloy can detect copper particles of 100μm, with an SNR of 1.65. This shows that the structure designed in this paper can detect copper particles below 100 μm, as shown in Figure 12b. The results of the increasing SNR of iron particles and copper particles of different sizes are shown in Table 1 below.

Table 1 shows the calculated SNR of particles with different sizes which were detected with different sensors. For 30 μm iron particles, the SNR of the sensor with two permalloy structures has increased by 47%. For 100 μm copper particles, the SNR has increased by 65%. The larger the particle size, the more obvious the increase in SNR value, and the detection capability of the sensor of this structure is significantly stronger than the sensor without permalloy and the sensor with a slice of permalloy. The detection limit of the dual permalloy structure sensor is 20 μm iron and 80 μm copper particles. Among the same type of sensors, this structure has the highest detection accuracy. The signal value is shown in the Figure 13 and Figure 14.

## 5. Conclusions

Table 2 shows the advantages and disadvantages of corresponding works. The comparison indicates that the detection accuracy from Hong [10] is relatively high, but it can only detect single pollutant types and has a moderate manufacturability. The detection accuracy of Du’s work [12] is relatively high and the production is simple, but it can only detect single pollutants types. The detection accuracy of Wei’s work [17] is relatively low, and it can detect single pollutant types. Ren’s work [11] can detect ferromagnetic and non-ferromagnetic pollutants, but the detection accuracy is relatively low. The works of [14,16] can detect relatively high accuracy, but there is still an improved level. 

This paper designs a sensor for monitoring metal pollutants in oil. The sensor has a high-gradient magnetic field structure by adding two high-permeability materials. Through simulation and experimental verification, it is confirmed that this structure can improve the detection accuracy of the inductive sensor. The simulation respectively simulated the magnetic field distribution of the sensor unit when there is no permalloy, one piece of permalloy, and two pieces of permalloy, and the optimal structure of the sensor unit is determined. Then, comparative experiments were performed on iron particles and copper particles of different sizes to obtain the SNR and the increase in percentage. For ferromagnetic metals, it can successfully detect 20 μm iron particles, and for non-ferromagnetic metals, it can detect the inductance signal of 80 μm copper particles. This result is the highest detection accuracy in the single-plane coil sensor, which has a guiding role in the design of this type of sensor.

## Figures and Tables

**Figure 1 micromachines-12-00638-f001:**
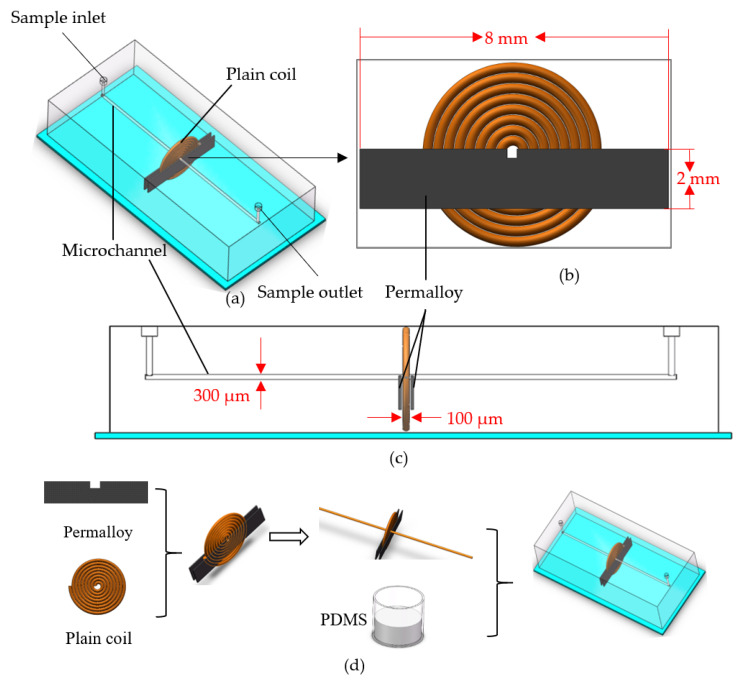
Sensor design diagram: (**a**) The overall design of the sensor; (**b**) sensing unit; (**c**) side view; (**d**) fabrication process.

**Figure 2 micromachines-12-00638-f002:**
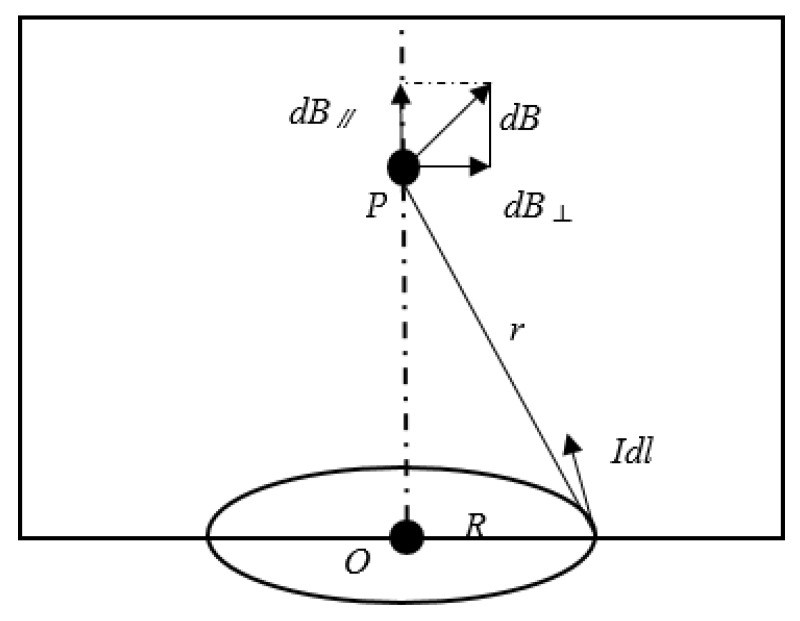
Magnetic field on the circular current-carrying conductor.

**Figure 3 micromachines-12-00638-f003:**
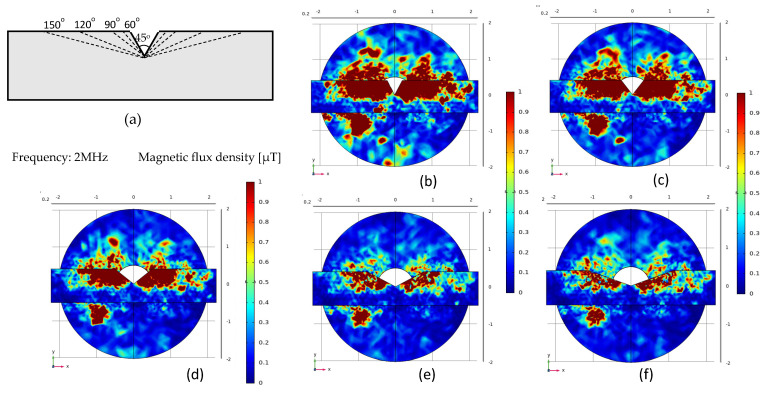
Comparison of simulation results of permalloy triangle notches: (**a**) Different permalloy structures; (**b**–**f**) Simulation results.

**Figure 4 micromachines-12-00638-f004:**
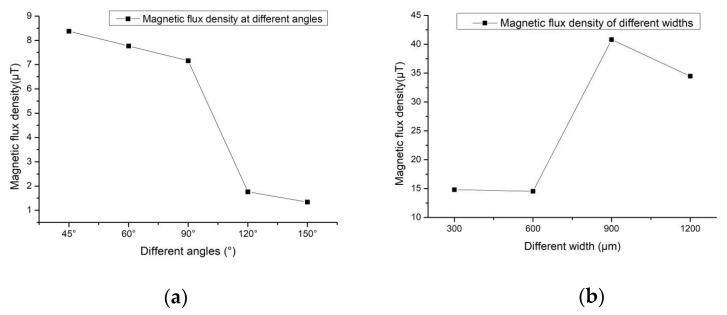
Results of different triangular and rectangular notches: (**a**) Corresponding results in Figure 3; (**b**) Corresponding results in Figure 5.

**Figure 5 micromachines-12-00638-f005:**
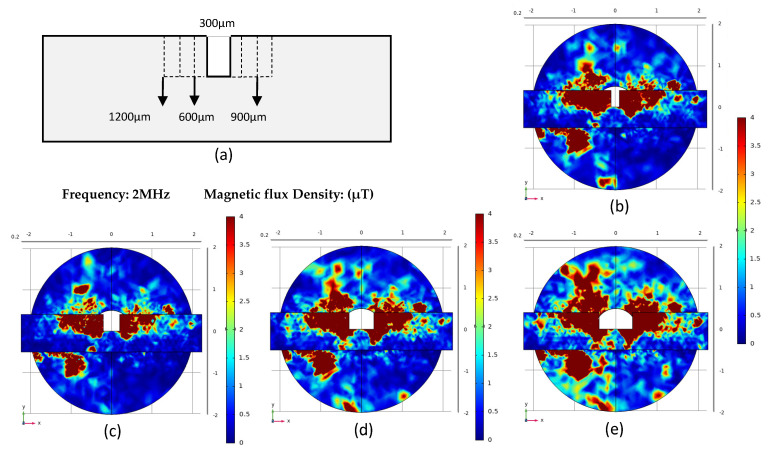
Comparison of simulation results of permalloy triangle notches: (**a**) Different permalloy structures; (**b**–**e**) Simulation results.

**Figure 6 micromachines-12-00638-f006:**
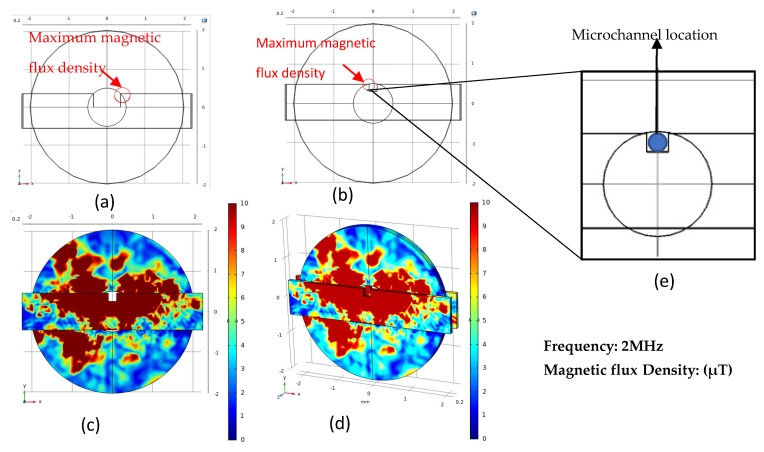
Permalloy structure with the highest magnetic flux density: (**a**) 1200μm wide rectangular slot, the maximum magnetic flux density is at the edge of the slot and inner hole of the coil; (**b**) A rectangular slot with a width of 300μm and a depth of 500μm, the maximum magnetic flux density is at the edge of the slot and inner hole of the coil; (**c**) Simulation results corresponding to the figure (**b**); (**d**) The result of two pieces of Permalloy in figure (**b**); (**e**) The Microchannel location in figure (**b**).

**Figure 7 micromachines-12-00638-f007:**
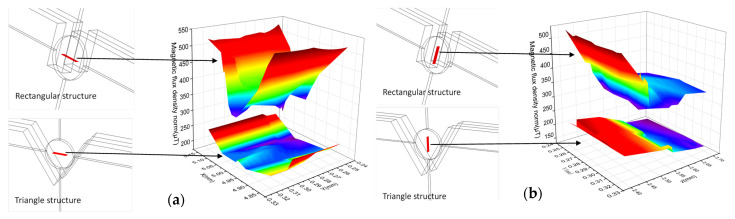
Distribution of magnetic fields in different directions in the induction area: (**a**) Horizontal magnetic field distribution; (**b**) Magnetic field distribution in the vertical direction.

**Figure 8 micromachines-12-00638-f008:**
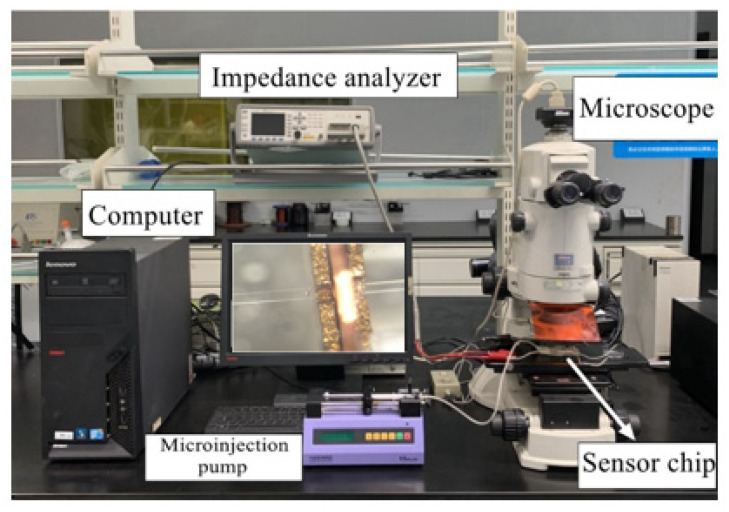
Experimental system.

**Figure 9 micromachines-12-00638-f009:**
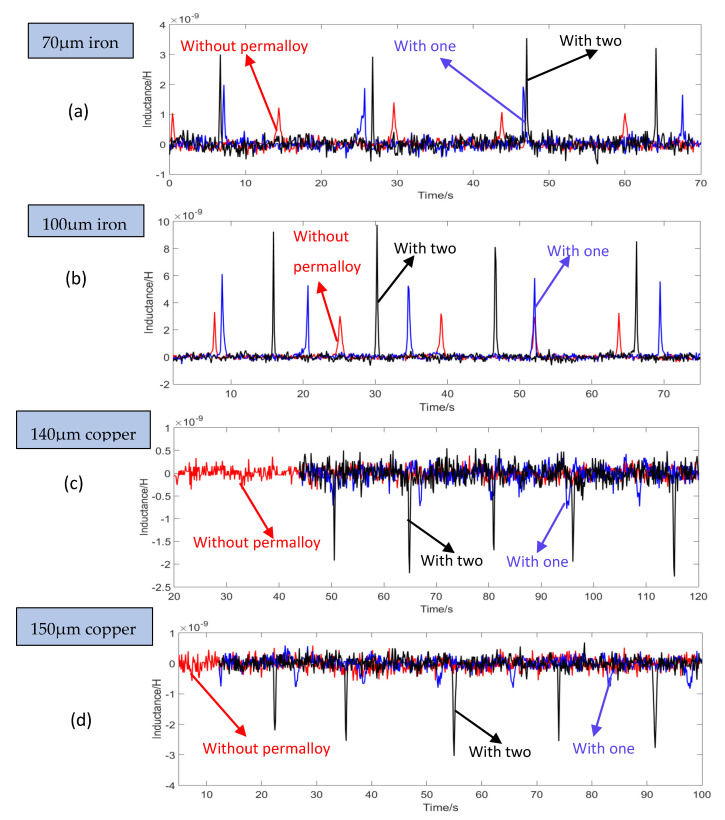
Signal value comparison: (**a**) 70 μm iron particles; (**b**) 100 μm iron particles; (**c**) 140 μm copper particles; (**d**) 150 μm copper particles.

**Figure 10 micromachines-12-00638-f010:**
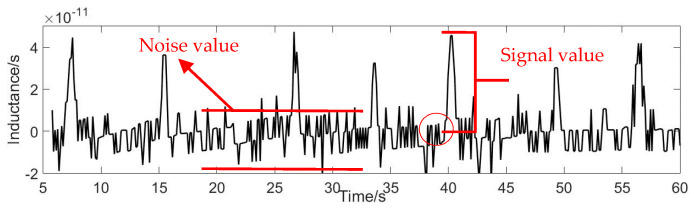
Signal value and noise value schematic diagram.

**Figure 11 micromachines-12-00638-f011:**
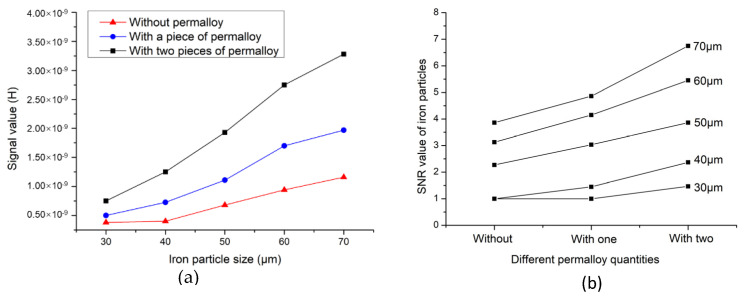
Iron particle detection results: (**a**) Average signal value of iron particles; (**b**) Average SNR value of iron particles.

**Figure 12 micromachines-12-00638-f012:**
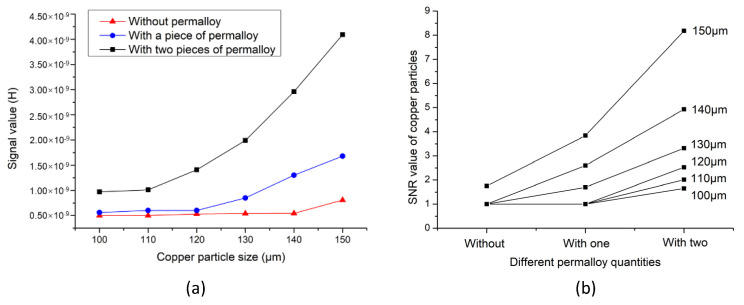
Copper particle detection results. (**a**) Average signal value of copper particles; (**b**) Average SNR value of copper particles.

**Figure 13 micromachines-12-00638-f013:**
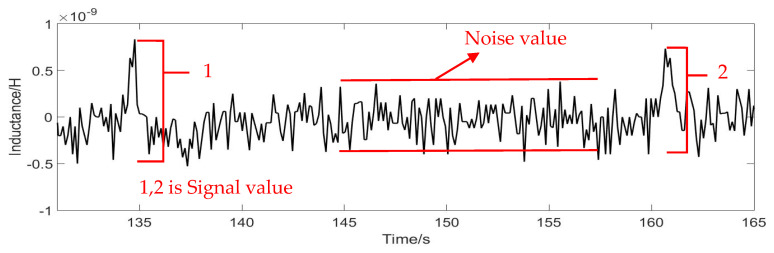
Signal of 20 μm iron particles.

**Figure 14 micromachines-12-00638-f014:**
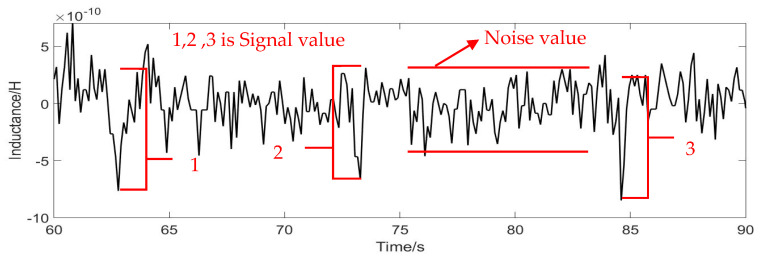
Signal of 80 μm copper particles.

**Table 1 micromachines-12-00638-t001:** SNR and added value in different sensors.

Iron (μm)	Without	With One	With Two	Coper (μm)	Without	With One	With Two
	SNR and Percentage Increase (%)				SNR and Percentage Increase (%)		
30	1	1/0%	1.47/47%	100	1	1/0%	1.65/65%
40	1	1.45/45%	2.37/137%	110	1	1/0%	2.01/101%
50	2.27	3.03/33.6%	3.86/70%	120	1	1/0%	2.53/153%
60	3.13	4.15/32.6%	5.45/74%	130	1	1.7/70%	3.32/232%
70	3.86	4.86/25.9%	6.75/74.9%	140	1	2.6/160%	4.93/393%
80	4.78	5.81/21.5%	8.56/79.1%	150	1.75	3.84/119%	8.18/367%

**Table 2 micromachines-12-00638-t002:** Comparison of other work.

The Works	Inductive Method	Detection of Pollutant Types	Detection Accuracy	Easy to Manufacture
[10]	Yes	Ferromagnetic	13 μm iron	Moderate
[12]	Yes	Ferromagnetic	55 μm iron	High
[17]	Yes	Ferromagnetic	81 μm iron	Moderate
[11]	Yes	Ferromagnetic Non-ferromagnetic	130 μm iron 230 μm copper	Moderate
[16]	Yes	Ferromagnetic Non-ferromagnetic	50 μm iron 130 μm copper	High
[14]	Yes	Ferromagnetic Non-ferromagnetic	30 μm iron 100 μm copper	High
This paper	Yes	Ferromagnetic Non-ferromagnetic	20 μm iron 80 μm copper	High

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
