# Peer review of "Design and Research of Inductive Oil Pollutant Detection Sensor Based on High Gradient Magnetic Field Structure"

_micromachines, 2021, doi:10.3390/mi12060638_

Round 1
Reviewer 1 Report
- Could the author include pictures of fabricated sensors and also real sensor test setup instead diagrams in Fig.7?
- what is the cross-section of the channel look like? can the authors make a fabrication process figure showing the fabrication of sensor in fig.1?
- In Fig. 8, the inductance of the signals are very small, given the very low signal value RLC meter to be detected, what potential noise source are the authors expecting? How do the authors set up the test environment to avoid/isolate the noise source?
- A legend in each subplot in fig. 8 should be added to show the particle size.
- what info is illustrated in fig 11 and fig.12, can they be combined? could the authors explain more on the signals in fig. 11 and fig. 12? which part in the figure is noise/signal?
Reviewer 2 Report
The authors in the present article demonstrated a prototype single coil inductive sensor for online detection and fault analysis of pollutants in hydraulic and lubricating oil systems. In spite of the nice methodology and new concept, the manuscript has many shortcomings. The authors should address the following comments:
- Many spelling mistakes are found through out the manuscript, such as “Biot-Saffor's law”, COMSOL Multiphasic etc. Kindly correct these.
- At many places, subscripts are not written in a proper manner. Please correct that.
- I do understand here Py is used as a magnetic flux concentrator, which eventually generates a gradient magnetic field in the air gap. In the “Detection principle and simulation analysis,” the effect of Py inclusion part is missing. Kindly include that numerical calculation part for both architectures.
- What would be the difference if we subtract equation (3) from equation (2)? What is the difference between B1 and B0?
- In COMSOL simulation, the generated field exhibits a very high value say, 76.7216
T (line # 209). It is a very high field and very difficult to control it. Please do all simulations with experimental set up parameters. Don’t use multiple digits after points. May be up to two digits is fine. - Please plot the magnetic field strength variation with height and horizontal axes. Add a comparative study based on two geometry configurations.
- It is also very difficult to understand whether measurements are performed in dynamic mode or in static mode. Kindly mention that part. Please also do discuss the amplitude analysis based on different flow rates, which is very important for that kind of this study.
- Why the signals exhibit opposite behavior as shown in Fig. 8(b) (with a piece of Py) and Fig. 8(c) (with two pieces of Py), respectively?
- It is also evident that Fig. 8(b) depicts higher signal amplitudes than Fig. 8(c) spectrum. If it is the case, then why “with a piece of Py” is not more advantageous, kindly address this point.
- Please refer the background noise for metallic and magnetic-metallic particles. Which filter is used to extract the noise data?
- What is the importance to show 20 ?m and 80 ?m particles signal data when any of the experiments was not performed for these sizes of the particles? Kindly clarify this point.
- Please add numerical equation for SNR calculation, which has been considered here.
- I would also like to know for a miniature device of this type of device, how the magnetic field gradient will affect?
- Kindly also add a comparative table with similar kind of works to show the importance of this work
- The fault experiment and analysis are missing here. Please add this part.
Author Response
Dear Reviewer:
Thank you for your letter and for the reviewers’ comments concerning our manuscript entitled “Design and research of inductive oil pollutant detection sensor based on high gradient magnetic field structure” (ID: micromachines-1220920). Those comments are all valuable and very helpful for revising and improving our paper, as well as the important guiding significance to our researches. We have studied comments carefully and have made correction which we hope meet with approval. Revised portion are marked in red in the paper. Please see the attachment.

Reviewer 3 Report
The authors propose a new inductive sensor design for a highly relevant application (the monitoring of metal pollutants in oil).
Please consider the following comments in order to improve the clarity of the paper.
Line 102
The microchannel diameter is 300 um and shall pass through the center of the coil. However, the coil has a diameter of only 100 um. This description is unclear.
Figure 1
Insert dimensions in the schematics for more clarity
Line 133
"Biot-Saffor"
Biot Savart
Please correct
Figure 2 has a poor resolution. Part of the text cannot be read. For instance dB??
All the variables written with subscripts in the formulae shall be written in text as well with a subscript. For instance B0 instead of B0.
Figure 3
Please write "Magnetic flux density [T]" instead of "Magnetic flux"
Please write the unit on the color bar
It is unclear why a frequency value of 2 MHz is indicated. This is not explained in the methodology. The physical analysis and the equations are magnetostatic equations.
The FEM simulation parameters shall be given with more details.
There is no interpretation of the magnetic flux density distribution on the planar coil's surface.
The authors mention several times the generation of a gradient within the channel in the center. They should give a 2D distribution of this gradient.
Line 172
Fig. 3. Please follow the template. Refer to figures with a constant style.
Line 175
The value of more than 8T for the simulated magnetic flux density seems extremely high. Please refer to the literature to illustrate that such values are realistic.
Line 228
Please reformulate
Line 231, 236
shaken instead of shake
Line 233
"were prepared" instead of "was prepared"
Line 237
The authors should indicate the flow rate of the injection syringe.
Line 243
Figure 8
Line 250
10-10
Use superscript 10-10
Figure 8
Text on the axis is too small
Explain the duration of the experiment.
Author Response
Dear Editors and Reviewer:
Thank you for your letter and for the reviewers’ comments concerning our manuscript entitled “Design and research of inductive oil pollutant detection sensor based on high gradient magnetic field structure” (ID: micromachines-1220920). Those comments are all valuable and very helpful for revising and improving our paper, as well as the important guiding significance to our researches. We have studied comments carefully and have made correction which we hope meet with approval. Revised portion are marked in red in the paper. Please see the attachment.

Round 2
Reviewer 1 Report
Thanks to the authors for the revision and it is in better shape.
Reviewer 2 Report
Thank you for your careful revision of the manuscript and responses to my questions. I genuinely believe these changes were a great improvement to the quality of your manuscript. However, a very few mistakes are there, which must be corrected
- Please change the magnetic flux density unit 8.38 (?T) to 8.38 ? (Line # 197). Similar mistakes are also observed in lines #200 and #231, respectively.
- Few labeling is missing in Fig. 10. Figure 10 caption must be written in a descriptive way.
- Please increase the resolution of Figure 3 (especially (b-f)).
- A typo is found in line #157. Please do correct it
- Line # 274 – flowrate is 50 ?l/min
- Correct the unit writing style in line #293 and #294
- In Table 1 – please write “With one” instead of “Withone”
- Please rewrite the title of “Table 2” in the correct form.
Reviewer 3 Report
The authors have significantly improved the clarity of the paper.